# The Impact of COVID-19 Lockdowns in a Sample of Outpatients in a Mental Health Department of a Metropolitan Hospital in Milan

**DOI:** 10.3390/medicina58091274

**Published:** 2022-09-14

**Authors:** Mauro Percudani, Matteo Porcellana, Ilaria Di Bernardo, Ivan Limosani, Attilio Negri, Luigi Zerbinati, Carla Morganti

**Affiliations:** 1Department of Mental Health and Addiction Services, Niguarda Hospital, Piazza Ospedale Maggiore 3, 20162 Milan, Italy; 2Department of Mental Health and Addiction Services, ASST Santi Paolo e Carlo, 20142 Milan, Italy; 3Institute of Psychiatry, Department of Biomedical and Specialty Surgical Sciences, University of Ferrara, 44121 Ferrara, Italy

**Keywords:** trauma spectrum, quarantine, psychiatric disorders, pandemic, coronavirus disease 2019

## Abstract

*Background and Objectives*: Italy was the first country in Europe to face the coronavirus 2019 (COVID-19) pandemic and its consequences, which led to two phases of severe restrictions for its population. This study aims to estimate the connections between the trauma of the COVID-19 emergency and the clinical features of a sample of outpatients in a Milan Community Mental Health setting, comparing the first (April 2020) and second lockdowns (November 2020). *Materials and Methods*: The sample included 116 consecutive outpatients recruited in April 2020 and 116 in November 2020. The subjects were evaluated with Clinical Global Impression Severity (CGI-S), Brief Psychiatric Rating Scale (BPRS-18), and Impact of Event Scale-Revised (IES-R). *Results*: The IES-R identified 47.4% participants in April and 50% in November with clinical scores over the cut-off. The network analysis of BPRS-18 and IES-R depicted the connection among different symptoms; in April, Unusual Thought Content, Anxiety, and Somatic Concern represented the most central items, and the strongest connections were found between Uncooperativeness and Hostility, Blunted Affect and Emotional Withdrawal, and IES-Intrusion and IES-Arousal. In the November group, the most central items were represented by Conceptual Disorganization and Emotional Withdrawal, whereas the strongest connections were found between IES-Arousal and IES-Intrusion, Excitement and Grandiosity, and Unusual Thought Content and Conceptual Disorganization. *Conclusions*: Our findings show continued high distress levels and increased psychological burdens during the second phase of restrictions; this could be described as “pandemic fatigue”, a general psychological weariness due to pandemic-related restrictions, as well as a lack of motivation to comply with them. As mental health professionals, our mission during these difficult times has been to keep community psychiatry services accessible, with particular regard to vulnerable and marginalized populations.

## 1. Introduction

The new coronavirus (SARS-CoV-2) rise represents an unprecedented challenge for public health worldwide [1].

On 30 January 2020, the International Health Regulations (2005) Emergency Committee declared the COVID-19 outbreak a Public Health Emergency of International Concern [2]. After its spread in most parts of the world, on 11 March 2020, the World Health Organization (WHO) declared the COVID-19 emergency a global pandemic [3].

The psychosocial burden associated with the spread of the pandemic, an extraordinary and unprecedented worldwide emergency, was expected to have a significant impact also on mental health, with potentially disruptive long-term effects [4]. 

The COVID-19 pandemic has exposed health workers to several risk factors impacting their wellbeing. Emerging studies highlight that the pandemic may create psychosocial risks also to community health workers (CHWs) that are poised to play a pivotal role in fighting the emergency [5]. The majority of the research in this field involved health workers in secondary care, while studies on psychosocial risks to CHWs are limited. Only a few studies have investigated the psychosocial response of CHWs during the pandemic. Ballard et al. [5] suggested an investment in the community health system and targeted actions to achieve specific goals in different steps of the pandemic. 

Franklin and Gkiouleka [6] carried out an interesting scoping review of psychosocial risks to health workers during the pandemic period. The review was conducted on global peer-reviewed literature, published between 1 January and 26 October 2020. An analysis of the extracted data found psychosocial risks related to four elements: personal protective equipment, job content, work organization, and social context. Moreover, women health workers and nurses showed worst health outcomes.

Ranieri et al. [7], in a cross-sectional study, based on two data detections (March 2020 and September 2020), described the post-traumatic stress disorder risk in healthcare workers, detecting the relationship between distress experience and personality dimensions in the Italian COVID-19 outbreak. In a short time, the impact was relevant, and protracted exposure to the stressors was related to personality traits. 

Although recent evidence reports sporadic cases of SARS-CoV-2 infections in the European Union since late 2019, Italy was the first EU country to officially record COVID-19 on its territory and to deal with the risks of its spread. In response to the exponential growth of cases recorded in Northern Italy in the last part of February 2020, the Italian Government implemented extraordinary measures to limit viral transmission in the last part of winter 2020, declaring a national lockdown for working and social activities, which were not considered essential, excluding healthcare, food distribution, and police forces. On 24 February 2020, there was the closure of the first schools in Milan, Lombardy, due to the risk of SARS-CoV-2 propagation. After a few weeks, travel to other EU countries was partially banned; this was followed on 9 March by banning gatherings of noncohabitating individuals.

Thus, the great majority of the Italian population was required to stay in their homes, refraining from public and social activities, including education, work, and family gatherings. This lasted for approximately two months. While the infection rate was steadily decreasing as an expected result of the restrictions, some of the measures were lifted starting on 4 May. During the following months, Italians were able to spend their summer holidays with almost no social restrictions; this was eventually followed by a new slow rise in SARS-CoV-2 infections, which peaked in late October 2020. The Italian Government responded with further, less stringent measures, known as the “second lockdown”. Indeed, in this phase, Italian institutions decided for a stratification of risk areas, with growing restriction rates from less prevalent areas (“white”/“yellow” regions) to high-risk zones (“orange”/“red” regions). Nevertheless, restrictions even in “red” areas were more flexible than during the spring; for instance, schools and several commercial activities were allowed to remain open, and individuals could perform open-air physical activities.

Several studies aimed to estimate the psychological distress related to the pandemic and subsequent lockdown, as well as to identify risk and/or protective factors among the general Italian population.

Mazza and colleagues [8] were among the first to administer an online survey to 2766 participants (18–22 March 2020); multivariate ordinal logistic regression showed a statistically significant association between higher levels of psychological distress and female gender, negative affectivity, and apathy. Depression and stress levels were significantly higher for those who had a relative or a friend diagnosed with COVID-19. Furthermore, subjects with a positive anamnesis for medical issues or stressful conditions were more likely to develop depression and anxiety. Anxiety and stress levels were also exacerbated by having a relative diagnosed with COVID-19, as well as by a younger age and by the necessity of leaving home to work (i.e., “essential” workers).

Moccia et al. [9] investigated a sample of 500 subjects, finding that 38% of them were experiencing some form of psychological distress. Moreover, levels of mental health burden were influenced by both temperament and affective features.

Tommasi et al. [10] carried out an internet survey aiming to report the impact of the March–April lockdown on a sample of 418 subjects. The results show that while Government measures to avoid infections were carefully followed by the majority of the sample, a significant reverberation on physical and mental health was commonly perceived; 43% of the subjects reported physical symptoms (e.g., migraine, sleep disorders, asthenia, and attention deficits). Anxiety levels were three times greater than the period before the pandemic, and low mood was reported in 30% of males and 41% of females.

Rossi et al. [11] indicated that 37% of the participants in a sample of 18,147 subjects had signs/symptoms suggestive of post-traumatic stress, while high levels of anxiety, perceived stress, sleep problems, and adjustment disorders were reported by 21–23% of individuals.

Quality of life and post-traumatic stress disorder (PTSD) symptoms among the general population in April 2020 were also investigated by Bonichini and Tremolada [12]. Among the 1839 anonymous volunteers, 23.5% of them had an Impact of Event Scale-Revised (IES-R) score higher than 33, whereas the most referred emotions were anxiety, impotence, boredom, and low mood.

Few studies have investigated the impact of the pandemic and related restrictions on psychiatric patients. In April 2020, we performed preliminary research to evaluate the impact of the COVID-19 emergency on 140 consecutive outpatients recruited in a community mental health clinic [13]. We reported a considerable proportion of distress measured by IES-R (32.1% moderate and 26.4% severe); this suggests the value and significance of an accurate assessment and monitoring of mental health patients’ conditions, both from a psychopathological and medical perspective.

The aim of this study is to extend our analysis by describing the consequences of the November lockdown, highlighting similarities and potential differences compared to the first one. Lombardy, where our study was conducted, was defined as a “red” zone (i.e., high-risk area) on 6 November. As we previously pointed out, restrictive measures were less rigorous in autumn 2020 compared to the previous lockdown. Considering these elements, it was conceivable that people suffering from pre-existing mental health issues would potentially suffer less during the second “light lockdown” than in the initial pandemic period.

## 2. Materials and Methods

### 2.1. Participants and Procedures

Our research study is naturalistic, exploratory, and descriptive.

In total, 232 subjects were recruited; 116 were selected between 5 April and 9 April 2020, while the other half was recruited over the second lockdown period between 9 November and 20 November 2020. All subjects were outpatients under psychiatric treatment at the Community Mental Health Service of the ASST Great Metropolitan Hospital Niguarda of Milan.

Inclusion criteria were age between 18 and 75, diagnosis of neurotic, stress related, and somatoform syndromes (F40-48), or affective syndromes (F30-39), or schizophrenia, schizotypal, and delusional disorders (F20-29), or personality disorders (F60-69) according to ICD-10.

Exclusion criteria were severe systemic or neurological illnesses, inability to give consent, or to perform self-report scales.

Informed consent was given by all participants; the Ethical Committee of Milan—Area 3 approved the research protocol (n. 360-24062020).

The research was conducted in compliance with the Declaration of Helsinki.

### 2.2. Survey Instrument

A structured interview was used to collect sociodemographic and clinical data.

Clinical Global Impression Severity of Illness (CGI-S) [14] as well as the 18-item Brief Psychiatric Rating Scale (BPRS-18) [15,16] assessed patients’ clinical conditions. 

Physician’s evaluation of patients’ current clinical state was reported by the CGI-S, considering a time span of a week before the evaluation; score ranged from 1 (normal, not at all ill) to 7 (among the most extremely ill patients).

The BPRS-18 is considered one of the most reliable quantitative scales for the measurement of psychiatric symptom severity and symptom evolution. Widely used in clinical and psychopharmacological trials, it provides a score ranging from 1 (not present) to 7 (extremely severe).

Similar to previous studies on the topic [12,13], we used the IES-R [17] to assess psychological distress; this 22-item self-report scale evaluated traumatic events related to subjective distress during the previous week, ranking it on a 5-point scale from 0 (not at all) to 4 (extremely). Maximum score is 88 (worst post-traumatic stress state). Cut-off for clinically relevant post-traumatic distress symptoms was set at 33 [18]. According to the fourth edition of the *Diagnostic and Statistical Manual of Mental Disorders* (DSM-IV) criteria for PTSD, IES-R assesses avoidance (the tendency to avoid thoughts or reminders about the trauma), intrusion (difficulty in wakefulness, dissociative thoughts, and reminiscences of the incident), and hyperarousal (irritation, anger, and insomnia) with three subscales.

### 2.3. Statistical Analysis

Statistical Package for the Social Science (SPSS) -20 package was used to perform statistical analysis. Continuous variables were reported as mean +/− standard deviation (SD), while categorical variables were showed as frequencies (%). Differences in clinical and sociodemographic dimensions in the two samples were explored using t-test or Chi-Square when appropriate.

A network analysis [19] was implemented in order to assess the association of BPRS items with IES-R subscales. The network was estimated according to a Graphical Gaussian Model (GGM), where the edges (the connections between the symptoms) represent conditional dependence relationships among the nodes (representing symptoms). The magnitude of the association is shown by the thickness of the edges. In other words, each edge connecting two nodes represents the unique shared covariance between two variables, after adjusting for all other nodes in the network. The R-package qgraph was employed to visualize the network structure. The Graphical Least Absolute Shrinkage and Selection Operator (GLASSO) procedure was used to highlight the strongest sets of connections and to obtain a sparse network [20,21].

## 3. Results

### 3.1. Participant Characteristics

Our sample consisted of 232 participants with a mean age of 51.36 years (range 24–75, sd 11.59). There were 137 females (59.1%) and 95 males (40.9%). The sociodemographic and diagnostic features among subgroups are summarized in Table 1.

### 3.2. Results of the Clinical and Stress Scales

We did not find any statistically significant difference between the two subgroups with regard to IES-R domains, while a higher intensity of symptoms was recorded for the November subgroup according to the CGI-S (t = −4.39, df = 250, *p* < 0.001, and Cohen’s d = −0.55) and BPRS total score (t = −2.65, df = 248, *p* < 0.01, and Cohen’s d= −0.33), as shown in Table 2.

### 3.3. Network Analysis

The networks of BPRS and IES-R dimensions are reported in the graphs of Figure 1 and Figure 2, which outline the connections among different psychopathological features in April and November 2020, respectively.

In April, Unusual Thought Content, Anxiety, and Somatic Concern represented as the most central items. The strongest edges were found between Uncooperativeness and Hostility (0.58), Blunted Affect and Emotional Withdrawal (0.41), and IES-Intrusion and IES-Arousal (0.53) (Table 3, Figure 3).

The November group displayed a much sparser network with fewer connections; the most central items are represented by Conceptual Disorganization and Emotional Withdrawal; a different group of symptoms is represented by symptoms of Suspiciousness, Hostility, Uncooperativeness, and Hallucinatory Behavior. The strongest edges were found between IES-Arousal and IES-Intrusion (0.62), Excitement and Grandiosity (0.51), and Unusual Thought Content and Conceptual Disorganization (0.4) (Table 4, Figure 4). Symptoms of disorientation, tension, guilt feelings, motor retardation, and avoidance did not connect to any node.

## 4. Discussion

To our knowledge, no previous research has explored the effect of the two consecutive COVID-19 lockdowns in a sample of psychiatric outpatients in Italy.

Two online surveys by Moradian et al. [22] investigated both variations and parallels between the two lockdowns (spring 2020 and autumn 2020) with regard to mental health and safety behavior in the general population. The investigators reported an increase in fear, generalized anxiety, low mood, and distress in the latter period, as well as less adherence to restrictions and safety measures; furthermore, an increased psychological burden and level of depression were recorded during the autumn lockdown. These findings have been interpreted as a new phenomenon named “pandemic fatigue”, i.e., a general psychological tiredness due to pandemic-related restrictions and lack of motivation to follow such rules. 

Our results show that a considerable percentage of patients manifested distress symptoms quantified by IES-R, both in April (47.4%) and November (50%) and that the increased prevalence of IES-R clinically relevant cases persisted during the second lockdown, supporting the concept of “pandemic fatigue” as described by Moradian and colleagues [22].

Moreover, during the second lockdown, patients reported a higher intensity of symptoms according to CGI-S and BPRS total scores in comparison with the early pandemic phase.

The available literature agrees on the higher susceptibility of psychiatric patients both to SARS-CoV-2 infection and its complications, as well as to detrimental consequences in terms of mental health, leading to self-isolation and abrupt discontinuation of their regular psychiatric care, including pharmacological treatment [23,24,25]. Indeed, the lack of awareness of these subjects regarding the risks of transmission of SARS-CoV-2, may lead to a lower compliance with COVID-19 prevention procedures, such as social distancing, strict hygiene, and isolation of positive cases, among others.

Regarding our network analysis, the results show that Unusual Thought Content, Anxiety, and Somatic Concern represented the most central symptoms in April, while in November, Conceptual Disorganization, Emotional Withdrawal, and Suspiciousness were prevalent.

The abrupt and unprecedented social restrictions introduced during the first lockdown were mandatory, in order to protect vulnerable populations and prevent hospital capacity overload [26]. As a consequence, normal habits of the Italian population suddenly changed, having detrimental consequences on several aspects of everyday life, including mental health treatments [27].

A number of studies showed an increased prevalence of depression- and anxiety-related symptoms, psychological distress, and COVID-19-associated preoccupations during the first lockdown [1,28,29,30,31]. SARS-CoV-2 has been presented as an extremely dangerous pathogen causing a sense of danger and uncertainty among both psychiatric patients and the general population. Consequently, the core symptoms identified during the first lockdown in our research can be explained by the propagation of a previously undiscovered fatal virus and related fears.

As expected, in April 2020, the main psychopathological dimension reported was related to anxiety. According to the existing literature [32], during the COVID-19 pandemic, such symptoms as well as a higher risk of depression were positively correlated with low-income populations and with the exposure to further stressors; it is known that psychiatric patients may be considered a vulnerable population [33] for several reasons. In fact, the psychiatric population may be more likely to contract SARS-CoV-2, as well as to experience higher difficulties to be tested and treated, leading such patients to a higher probability to develop both physical and psychological complications [34]. Hence, in line with the expectations, the anxious psychopathological spectrum highlighted in our sample identified the core symptoms exhibited during the first pandemic period.

Compared to the first lockdown, in autumn 2020, the Italian Government issued a number of less strict rules; thus, we hypothesized that people with pre-existing mental health issues would potentially suffer less during the second “light lockdown”. 

Nevertheless, the present study shows a persistently high psychopathological burden in the second lockdown, despite fewer restrictions and the specific reorganization of the Mental Health Service, which is consistent with previous findings in a general German population sample [22]. Specifically, in November 2020, the core symptoms were represented by Conceptual Disorganization, Emotional Withdrawal, and Suspiciousness, identifying a predominance of the psychotic spectrum in contrast with the first lockdown, which was characterized by anxious symptoms. This shift to psychotic spectrum symptoms may be explained by “pandemic fatigue” [22,35].

Finally, prolonged stress can easily induce PTSD with secondary psychotic features (PTSD-SP), which may be considered a discrete entity of PTSD with peculiar risk factors increasing its prevalence and determining its course [36], such as a pre-existing depression disorder or substance use disorder.

Consequently, in subjects with psychiatric disorders, the onset of a PTSD-SP in prolonged stressing situations may be frequent compared to the general population. 

The study is strengthened by solid methodology in the exploration of the relationship between different symptom domains. Nonetheless, it presents several limitations, a fact which precludes generalizations. 

First, the sample consisted of two different groups; although they did not differ socio-demographically or by diagnosis, a better option would have been to simply reassess the same participants as a follow-up procedure. The differences showed in the network structures can be thus attributed either to differences among patients or as a consequence of the different time of assessment. 

Second, the sample size of the study is limited and allows only for exploratory considerations. Unfortunately, the pandemic emergency and the consequent urge to reorganize access to mental health services made it difficult to recruit a wider number of patients or to reassess them properly. 

Finally, PTSD symptoms were assessed with a self-reported scale, and no psychiatric structured clinical interview was used.

## 5. Conclusions

Lombardy is the most densely populated region in Italy, with a population of around 10 million people; the intense circulation of people and goods led the region to be most affected by the COVID-19 pandemic and its consequences in Italy. Our findings show prolonged high levels of distress and increased levels of psychological burden in the second lockdown, which may be interpreted as pandemic fatigue among mental health patients. This suggests the importance of strict monitoring of such population conditions, both from psychopathological and general health perspectives.

As the COVID-19 pandemic still rages worldwide, exceptional care should be provided to the most vulnerable subjects, including those referred to community psychiatric clinics; this is not only a moral imperative, but also a public health responsibility [37,38].

## Figures and Tables

**Figure 1 medicina-58-01274-f001:**
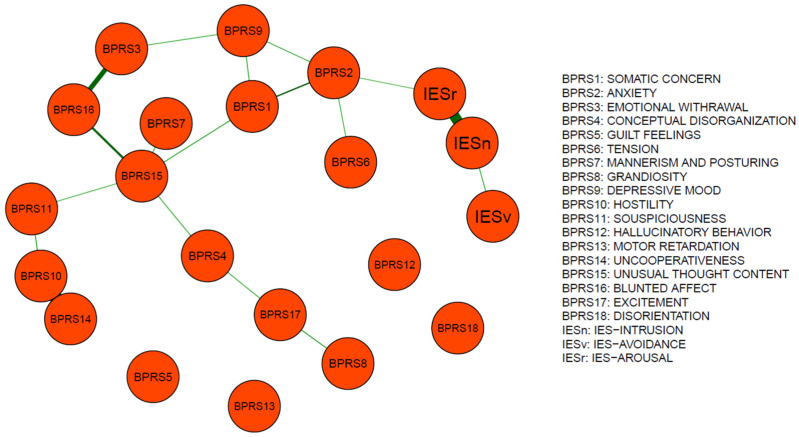
Network structure in the April sample. Edge weights reflect the connection strength of BPRS factors with IES-R dimensions in the April network.

**Figure 2 medicina-58-01274-f002:**
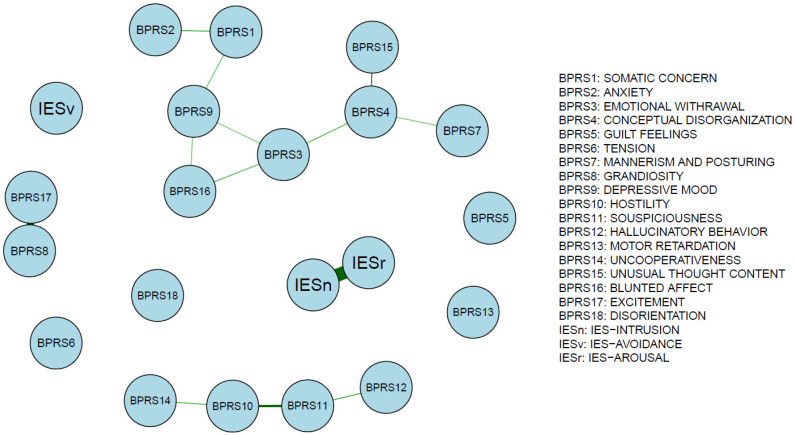
Network structure in the November sample. Edge weights reflect the connection strength of BPRS factors with IES-R dimensions in the November network.

**Figure 3 medicina-58-01274-f003:**
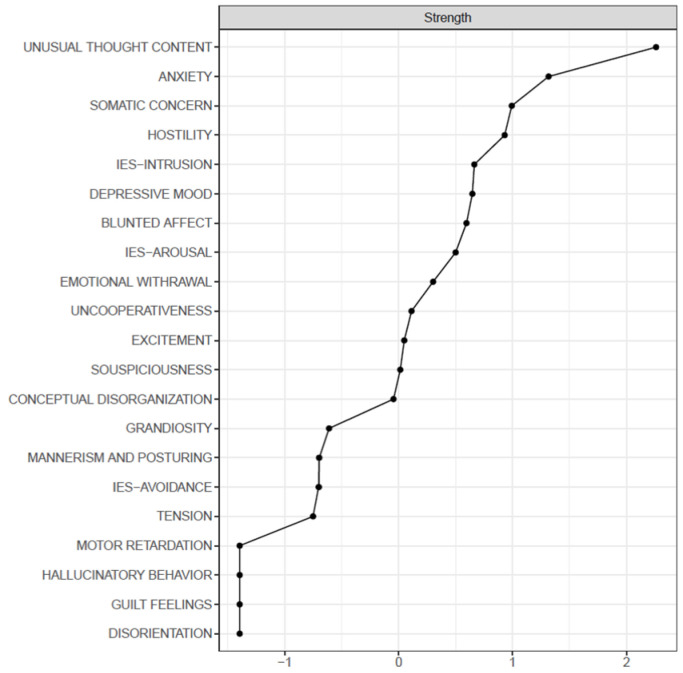
Centrality plot (strength) of the April sample.

**Figure 4 medicina-58-01274-f004:**
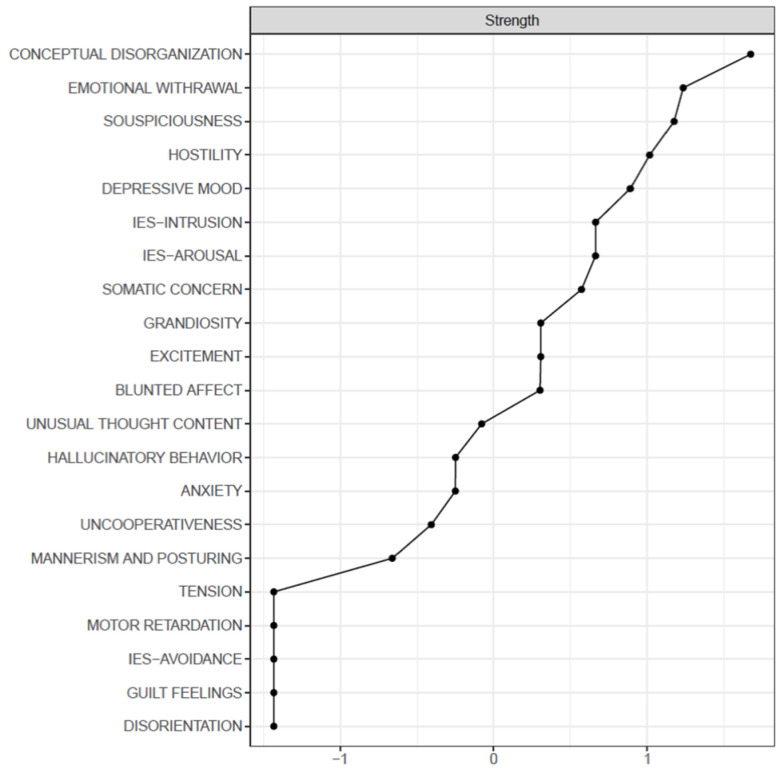
Centrality plot (strength) of the November sample.

**Table 1 medicina-58-01274-t001:** Characteristics of participants in subgroups.

	April (*n* = 116)		November (*n* = 116)		
**Age**	50.68 ± 11.59		51.39 ± 11.97		t = −0.47, df = 250, *p* > 0.05, Cohen’s d = −0.05
**Civil Status**					
unmarried	66	52.4%	66	52.4%	χ^2^ = 2.82, df = 3, *p* > 0.05
married	35	27.8%	41	32.5%	
separated/divorced	23	18.3%	15	11.9%	
widowed	2	1.6%	4	3.2%	
**Education**					
primary	2	1.6%	2	1.6%	χ^2^ = 7.72, df = 3, *p* > 0.05
middle	32	25.4%	42	33.3%	
high school	63	50.0%	69	54.8%	
graduate/postgraduate	29	23.0%	13	10.3%	
**Living Situation**					
alone	47	37.3%	39	31.0%	χ^2^ = 1.54, df = 3, *p* > 0.05
partner	51	40.5%	52	41.3%	
relatives	23	18.3%	29	23.0%	
other	5	4.0%	6	4.8%	
**Work Status**					
unemployed	16	12.7%	20	15.9%	χ^2^ = 9.83, df = 3, *p* < 0.05
employed	65	51.6%	54	42.9%	
retired	7	5.6%	16	12.7%	
student/housewife	9	7.1%	2	1.6%	
invalid	29	23.0%	34	27.0%	
**Psychiatric Diagnosis**					
schizophrenia	38	30.2%	43	34.1%	χ^2^ = 7.93, df = 4, *p* > 0.05
bipolar disorder	33	26.2%	19	15.1%	
depressive disorder	22	17.5%	24	19.0%	
anxiety disorder	3	2.4%	10	7.9%	
personality disorder	30	23.8%	30	23.8%	

**Table 2 medicina-58-01274-t002:** Psychopathological dimensions and severity in subgroups.

	April	November	Statistics
CGI-S	3.38 ± 0.90	3.83 ± 0.72	t = −4.39, df = 250, *p* < 0.001,Cohen’s d = −0.55
IES-R Intrusion	11.31 ± 6.24	11.89 ± 6.50	t = −0.71, df = 249, *p* > 0.05, Cohen’s d = −0.09
IES-R Avoidance	10.9 ± 6.68	10.72 ± 6.02	t = 0.22, df = 249, *p* > 0.05, Cohen’s d = 0.02
IES-R Arousal	9.32 ± 5.37	9.18 ± 5.18	t = 0.21, df = 249, *p* > 0.05, Cohen’s d = 0.02
IES-R Total score	31.44 ± 15.44	31.7 ± 14.67	t = −0.14, df = 249, *p* > 0.05, Cohen’s d = −0.01
**PTSD cases**			
Cases (IES-R < 33)	61 52.6%	58 50.0%	χ^2^ = 0.15, df = 1, *p* > 0.05
Cases (IES-R ≥ 33)	55 47.4%	58 50.0%	
BPRS Total	37.69 ± 8.36	40.84 ± 10.28	t = −2.65, df = 248, *p* < 0.01, Cohen’s d = −0.33

**Table 3 medicina-58-01274-t003:** Weighted adjacency matrix of the April sample.

	Somatic Concern	Anxiety	Emotional Withdrawal	Conceptual Disorganization	Guilt Feelings	Tension	Mannerism And.Posturing	Grandiosity	Depressive Mood	Hostility	Suspiciousness	Hallucinatory Behavior	Motor Retardation	Uncooperativeness	Unusual Thought Content	Blunted Affect	Excitement	Disorientation	IES-INTRUSION	IES-AVOIDANCE	IES-AROUSAL
Somatic Concern		0.34							0.29						0.29						
Anxiety	0.34					0.25			0.25												0.2
Emotional Withdrawal									0.24							0.41					
Conceptual Disorganization															0.27		0.25				
Guilt Feelings																					
Tension		0.25																			
Mannerism And Posturing															0.27						
Grandiosity																	0.3				
Depressive Mood	0.29	0.25	0.24																		
Hostility											0.32			0.58							
Suspiciousness										0.32					0.23						
Hallucinatory Behavior																					
Motor Retardation																					
Uncooperativeness										0.58											
Unusual Thought Content	0.29			0.27			0.27				0.23					0.36					
Blunted Affect			0.41												0.36						
Excitement				0.25				0.3													
Disorientation																					
IES-INTRUSION																				0.27	0.53
IES-AVOIDANCE																			0.27		
IES-AROUSAL		0.2																	0.53		

**Table 4 medicina-58-01274-t004:** Weighted adjacency matrix of the November sample.

	Somatic Concern	Anxiety	Emotional Withdrawal	Conceptual Disorganization	Guilt Feelings	Tension	Mannerism and Posturing	Grandiosity	Depressive Mood	Hostility	Suspiciousness	Hallucinatory Behavior	Motor Retardation	Uncooperativeness	Unusual Thought Content	Blunted Affect	Excitement	Disorientation	IES-INTRUSION	IES-AVOIDANCE	IES-AROUSAL
Somatic Concern		0.35							0.24												
Anxiety	0.35																				
Emotional Withdrawal				0.29					0.21							0.28					
Conceptual Disorganization			0.29				0.23								0.4						
Guilt Feelings																					
Tension																					
Mannerism and Posturing				0.23																	
Grandiosity																	0.51				
Depressive Mood	0.24		0.21													0.23					
Hostility											0.42			0.3							
Suspiciousness										0.42		0.35									
Hallucinatory Behavior											0.35										
Motor Retardation																					
Uncooperativeness										0.3											
Unusual Thought Content				0.4																	
Blunted Affect			0.28						0.23												
Excitement								0.51													
Disorientation																					
IES-INTRUSION																					0.62
IES-AVOIDANCE																					
IES-AROUSAL																			0.62		

## Data Availability

The data presented in this study are available on request from the corresponding author.

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
