# Peer review of "The Impact of COVID-19 Lockdowns in a Sample of Outpatients in a Mental Health Department of a Metropolitan Hospital in Milan"

_medicina, 2022, doi:10.3390/medicina58091274_

Round 1

Reviewer 1 Report

The design of the study do not permit to draw definitive conclusion about a causal relationship between the changes that have been observed and the  pandemic but is suggestive.

A comparison for subgroups of pathology could be interesting.

Author Response

Dear Reviewer, thank you for your suggestions.

We specified in the limitations of the study that, due to sample size, it could not draw definitive conclusions on the connection between the lockdowns and the changes observed; thus, our work only allows preliminary, exploratory considerations.

A comparison of the different diagnosis subgroups will be considered for further studies.

Reviewer 2 Report

I do not find this paper's findings significant and relevant in the terms of some serious scientific research.

If the authors wanted to do some longitudinal, i.e. follow up study, then they should prepare data for longer intervals (not just April and November the same year) and use the appropriate statistic analysis to better interpret data.

Author Response

Dear Reviewer, thank you for your suggestions.

As highlighted in the limitations, our study only allows exploratory consideration on the topic. The aim of our study was to provide a naturalistic, exploratory and descriptive research focused on possible similarities and differences between two unique periods in COVID-19 pandemic. Thus, it was not possible to consider longer intervals  to retrieve data.

Further studies may be prepared in order to assess long term consequences of this unprecedented phenomenon.

Reviewer 3 Report

I'm thankful to review this interesting manuscript.

The topic is relevant and the outcome of the study useful for future research.

Intrudction could be improved including more taylored papers as almost references are base on general population; could be better to discuss scientific evidence on healthcare workers because as paper was focused on mental health outpatients. Following, discussion paragraph should be integrated.

Statistical analyses and more elaboration data sound essential but clear and well explained.

The findings are relevant but should be argued better with exisitng literature. 

Suggested references for improvments:

Ballard MBancroft ENesbit J, et al Prioritising the role of community health workers in the COVID-19 response  

- Franklin P, Gkiouleka A. A Scoping Review of Psychosocial Risks to Health Workers during the Covid-19 Pandemic. International Journal of Environmental Research and Public Health. 2021; 18(5):2453. https://doi.org/10.3390/ijerph18052453

Ranieri, J., Guerra, F., Perilli, E., Passafiume, D., Maccarone, D., Ferri, C., & Di Giacomo, D. (2021). Prolonged COVID 19 Outbreak and Psychological Response of Nurses in Italian Healthcare System: Cross-Sectional Study. Frontiers in psychology12, 608413. https://doi-org.univaq.clas.cineca.it/10.3389/fpsyg.2021.608413

Author Response

Thank you for your review.

Suggested references have been included and briefly commented in the text.

Round 2

Reviewer 2 Report

Dear authors, as I said before, I do not find this paper's findings significant and relevant in the terms of some serious scientific research. That is why I do not think this paper is suitable for "Medicina". Therefore I propose you other journals.

However, if the editors accept this paper in the form of only"providing a naturalistic, exploratory and descriptive research ...", than you are not going to have a problem in publishing it here also

Reviewer 3 Report

Dear Authors 

the revised version of paper is suitable for acceptance.